# FAMMA: A Benchmark for Financial Multilingual Multimodal Question Answering

## Abstract

In this paper, we introduce FAMMA, an open-source benchmark for financial multilingual multimodal question answering (QA).[1] Our benchmark aims to evaluate the abilities of multimodal large language models (MLLMs) in answering questions that require advanced financial knowledge and sophisticated reasoning. It includes 1,758 meticulously collected question-answer pairs from university textbooks and exams, spanning 8 major subfields in finance including corporate finance, asset management, and financial engineering. Some of the QA pairs are written in Chinese or French, while a majority of them are in English. These questions are presented in a mixed format combining text and heterogeneous image types, such as charts, tables, and diagrams. We evaluate a range of state-of-the-art MLLMs on our benchmark, and our analysis shows that FAMMA poses a significant challenge for these models. Even advanced systems like GPT-4o and Claude-35-Sonnet achieve only 42% accuracy. Additionally, the open-source Qwen2-VL lags notably behind its proprietary counterparts. Lastly, we explore GPT o1-style reasoning chains to enhance the models' reasoning capabilities, which significantly improve error correction. Our FAMMA benchmark will facilitate future research to develop expert systems in financial QA. The code and data have been anonymously released at https://github.com/random2024GO/bench-script.

## 1 Introduction

Benchmarks have played a pivotal role in advancing AI research, particularly in the realm of large language models (LLMs) (Brown et al., 2020; OpenAI, 2023; Touvron et al., 2023; Jiang et al., 2023; 2024; Meta, 2024). Benchmarks have been helping researchers track the advancement of LLMs in a variety of capabilities, including general language understanding and knowledge acquisition (Wang et al., 2019; Hendrycks et al., 2021a; Zhou et al., 2023; Wang et al., 2024b), code generation (Chen et al., 2021a; Liu et al., 2023; Jimenez et al., 2024), mathematical reasoning (Cobbe et al., 2021; Hendrycks et al., 2021b), tool use (Yan et al., 2024; Srinivasan et al., 2023; Trivedi et al., 2024), and legal reasoning (Guha et al., 2023). Meanwhile, we have seen a scarcity of high-quality benchmarks in financial reasoning, an area where practitioners are eager to benefit from LLMs.

We envision that LLMs will have a broad and significant impact in the finance industry, enabling intelligent systems that can assist human experts in various tasks such as risk management and predictive analytics. Towards this goal, high-quality benchmarks are needed to track the capabilities of LLMs in understanding financial knowledge and answering complex financial questions. Unfortunately, existing benchmarks in this domain cannot fully reflect the nature of daily work of financial practitioners: they only have text-based questions; they are all in English; answering their questions only requires knowledge at a rudimentary to intermediate level (Hendrycks et al., 2021a; Chen et al., 2021b; Islam et al., 2023; Xie et al., 2024). For a detailed comparison, refer to section 2. However, financial practitioners often have to handle other data modalities, read documents in other languages, and use advanced knowledge. For example, traders often rely on charts to identify trading opportunities; financial analysts need to analyze documents that include many complicated tables; investors

---

[1]This name is made to honor Eugene Fama, a Nobel prize winner that is best known for his work on portfolio theory, asset pricing, and the efficient-market hypothesis. He is regarded as "the father of modern finance".

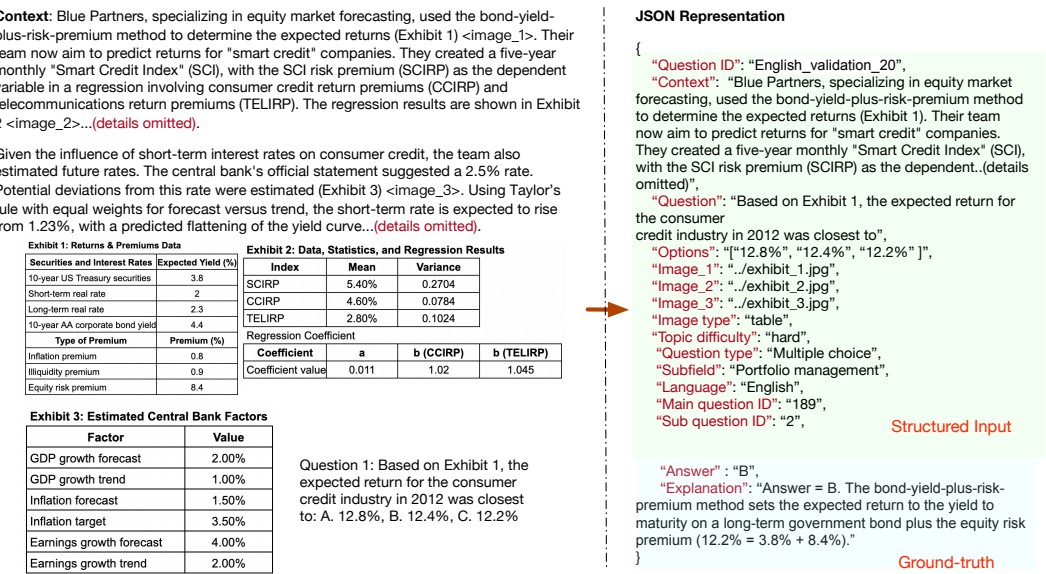

Figure 1: Sample question in portfolio management, classified as hard difficulty. Typically covered in master's courses. Prerequisites include master's-level knowledge of statistics and optimization theory. See the course description of Fin-504,505 in Princeton and outlines of portfolio management of CFA-Level III.

sometimes have to read earning reports in languages other than English; quantitative researchers often need to use advanced knowledge such as stochastic calculus to price financial contracts.

In this paper, we present FAMMA, an open-source benchmark for financial multilingual multimodal question answering (QA). Figure 2 displays three QA examples in our benchmark. Compared to existing benchmarks, FAMMA has a significantly better reflection of the real problems that financial practitioners address on a daily basis. This benchmark includes 1,758 meticulously collected question-answer pairs from university textbooks and exams, spanning 8 major subfields including corporate finance, asset management, and financial engineering. Answering these questions requires advanced knowledge such as factor models, option pricing, and asset allocation. A good portion of the QA pairs are written in Chinese or French, although a majority of them are in English. These questions are presented in a mixed format combining text and heterogeneous image types, such as charts, tables, and diagrams.

We evaluate 3 advanced proprietary MLMMs such as GPT-4o (OpenAI, 2023) and 1 top-ranked open-source MLMM—Qwen2-VL (Wang et al., 2024a). Our key findings are summarized below:

- FAMMA presents significant challenges: GPT-4o and Claude-35-Sonnet only achieve $42\%$ accuracy, notably lower than human performance $56\%$, indicating substantial room for improvement. In addition, there is a pronounced disparity in performance between Qwen2-VL and GPT-4o.

- Our analysis of 100 GPT-4o error cases reveals that 42.5% are due to domain knowledge gaps, while 27.5% involve ambiguous responses. This suggests that GPT-4o struggles with financial knowledge and at times generates imprecise answers despite correctly understanding the problem.

- We explore GPT o1-style reasoning chains to enhance the models' reasoning capabilities, significantly outperforming the RAG method in correcting errors on FAMMA, particularly in the categories of ambiguous answer generation and numerical inaccuracy.

## 2 RELATED WORK

The application of natural language technologies in finance dates back to the early 2000s, when sentiment analysis was used to analyze how media would impact stock market movements (Tetlock, 2007; Pang et al., 2008). Over recent years, the emergence of LLMs has inspired research in advancing financial industry with LLMs, including pretraining and fine-tuning LLMs with finance-related

**Context:** Clever Company and NY Patriot are competing manufacturing firms. Their financial statements are printed here.

**Balance Sheet of Clever Company**

| Assets | 2015 ($) | 2014 ($) |
|---|---|---|
| Current assets: | | |
| Cash | 13,862 | 16,339 |
| Net accounts receivable | 23,887 | 25,778 |
| Inventory | 54,867 | 43,287 |
| Total current assets | 92,616 | 85,404 |
| Fixed assets: | | |
| Plant, property, and equipment | 101,543 | 99,615 |
| Less: Accumulated depreciation | -34,331 | -31,957 |
| Net fixed assets | 67,212 | 67,658 |
| Prepaid expenses | 1,914 | 1,791 |
| Other assets | 13,052 | 13,138 |
| **Total assets** | **174,794** | **167,991** |
| **Liabilities and Equity** | 2015 ($) | 2014 ($) |
| (details omitted)... | | |
| Total current liabilities | 34,323 | 32,197 |
| Long-term debt | 22,036 | 22,036 |
| Total liabilities | 56,359 | 54,233 |
| Equity | | |
| Common stock | 38,000 | 38,000 |
| Paid-in capital | 12,000 | 12,000 |
| Retained earnings | 68,435 | 63,758 |
| Total equity | 118,435 | 113,758 |
| **Total liabilities and equity** | **174,794** | **167,991** |

**Balance Sheet of New York Patriot**

| Assets | 2015 ($) | 2014 ($) |
|---|---|---|
| Current assets: | | |
| Cash | 3,307 | 5,794 |
| Net accounts receivable | 22,133 | 26,177 |
| Inventory | 44,661 | 46,463 |
| Total current assets | 70,101 | 78,434 |
| Fixed assets: | | |
| Plant, property, and equipment | 31,116 | 31,842 |
| Less: Accumulated depreciation | -18,143 | -19,297 |
| Net fixed assets | 12,973 | 12,545 |
| Prepaid expenses | 688 | 763 |
| Other assets | 1,385 | 1,601 |
| **Total assets** | **85,147** | **93,343** |
| **Liabilities and Equity** | 2015 ($) | 2014 ($) |
| (details omitted)... | | |
| Total current liabilities | 13,910 | 19,672 |
| Equity | | |
| Common stock | 20,576 | 20,576 |
| Paid-in capital | 5,624 | 5,624 |
| Retained earnings | 46,164 | 48,598 |
| Less: Treasury stock | -1,127 | -1,127 |
| Total equity | 71,237 | 73,671 |
| **Total liabilities and equity** | **85,147** | **93,343** |

**Income Statement of Clever Company (2015)**

| Income: | 2015 ($) |
|---|---|
| Sales | 162,749 |
| Other income | 1,002 |
| Total income | 163,751 |
| Operating expenses | (details omitted) |
| Total expenses | 134,339 |
| Pretax earnings | 29,412 |
| Taxes | 14,890 |
| Net earnings | 14,522 |
| Dividends | 9,845 |
| Retained earnings | 4,677 |

**Income Statement of New York Patriot (2015)**

| Income: | 2015 ($) |
|---|---|
| Sales | 91,374 |
| Other income | 1,067 |
| Total income | 92,441 |
| Operating expenses | (details omitted) |
| Total expenses | 78,264 |
| Pretax earnings | 14,177 |
| Taxes | 6,838 |
| Net earnings | 7,339 |
| Dividends | 4,905 |
| Retained earnings | 2,434 |

Question 1: Which firm has the larger investment in current assets?
Question 2: Which firm is more likely to incur carrying costs?

(a) Sample question in financial statement analysis, classified as medium difficulty. Typically covered in master's courses. Prerequisites include senior undergraduate's-level or higher knowledge of accounting and corporate finance. See the course description of Fin-502 in Princeton and outlines of financial statement analysis of CFA-Level II.

**Context:** We have a three-period binomial model shown below. At time zero, we have a stock whose price per share we denote by $S_0$, a positive quantity known at time zero. At time one, the price per share of this stock will be one of two positive values, where the H and T standing for head and tail, respectively... (details omitted)

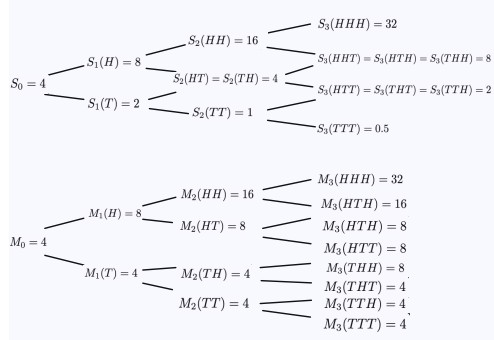

Question 1: Assume risk-neural probability for the up and down move are both 0.5, compute the conditional expectation of based on the information at time 1 under the risk neural measure $\hat{E}[S_2](H), \hat{E}[S_2](T)$

Question 2: Under the actual probability with the up and down move probability 2/3 and 1/3, consider the maximum-to-date process below $M_n = \max_{0 \le k \le n} S_k$, compute $\mathbb{E}_t[M3](TH), \mathbb{E}_t[M3](TT)$ and determine whether $M_n$ is Markov?

(b) Sample question in derivatives, classified as hard difficulty. Typically covered in master's courses. Prerequisites include master's level or higher knowledge of probability theory and stochastic calculus. See the course description of Fin-501,503 in Princeton and outlines of derivatives of CFA-Level III.

Figure 2: Sampled FAMMA examples from the other two subfields. The questions and images need expert-level knowledge to understand and reason. Samples in this figure are text truncated due to space.

text (Wu et al., 2023; Yang et al., 2023), improving sentiment analysis with LLMs (Konstantinidis et al., 2024; Inserte et al., 2024; Cao et al., 2024), and building chatbots that specialize in finance knowledge (Chase, 2022; Stratosphere-Technology, 2023; Xue et al., 2023; 2024).

Several existing benchmarks can be used to evaluate these modern models and systems, including FiQA (Maia et al., 2018), FinQA (Chen et al., 2021b), ConvFinQA (Chen et al., 2022), FinanceBench (Islam et al., 2023), and FinBen (Xie et al., 2024). However, these benchmarks cannot reflect the nature of real problems that financial practitioners have to deal with on a daily basis. In particular, their data only has text but not data of other modalities; their data is only in English; their questions only test knowledge at a rudimentary to intermediate level. The finance-related questions in MMMU (Yue et al., 2024) involve data of other modalities like tables and charts. However, this general-purpose benchmark covers multiple disciplines (e.g., art, business, science, humanities, etc), and thus has a very limited coverage on finance-related questions. Our FAMMA benchmark makes a unique and focused contribution to the community on top of existing benchmarks: it has a much broader coverage on subfields of finance; its data is in multiple languages and of multiple modalities; its questions test advanced knowledge.

## 3 THE FAMMA BENCHMARK

FAMMA provides comprehensive coverage across eight key subfields: alternative investments, corporate finance, derivatives, economics, equity, financial statement analysis, fixed income, and portfolio management. These topics closely align with those taught in elite academic programs, such as Princeton's Master in Finance, as well as professional certifications like the CFA program. The dataset consists of both multiple-choice (55.5%) and open questions (45.5%). Additionally, 70.4% of the questions feature single-image scenarios, while 29.6% involve multi-image scenarios. Notably, 99.5% of the questions are accompanied by explanations. The questions are distributed across three difficulty levels and three most widely used languages in the finance industry (eFinancialCa-

reers, 2022): English (78.8%), Chinese (14.4%), and French (6.8%). FAMMA is divided into a validation set and a test set. The validation set, useful for hyperparameter selection, contains 120 questions, while the test set comprises 1638 questions The overall subject coverage and statistics are shown in Table 1 while distribution of questions by languages and subfields are shown in Figure 3 and Figure 4, respectively. More detailed descriptive statistics can be found in Table 7 and Table 8 in Appendix A.

| Statistics | Numbers (%) |
|---|---|
| Total Questions | 1758 |
| * Multiple-choice | 976 (55.5%) |
| * Open | 782 (44.5%) |
| Difficulties | 608 / 438 / 712 |
| (Easy:Medium:Hard) | 34.6% : 24.9% : 40.5% |
| | # Tokens Avg |
| By Input and Output | |
| * Questions | 233.43 |
| * Explanation | 73.95 |
| By Splits | |
| * validation | 224.29 |
| * test | 234.12 |
| By Languages | |
| * English | 257.11 |
| * Chinese | 94.06 |
| * French | 254.28 |
| By Difficulty Levels | |
| * Easy | 136.62 |
| * Medium | 109.68 |
| * Hard | 375.25 |
| By Subfields | |
| * Alternative Investments | 473.33 |
| * Corporate Finance | 81.74 |
| * Derivatives | 277.47 |
| * Economics | 567.64 |
| * Equity | 243.15 |
| * Financial statement analysis | 49.67 |
| * Fixed income | 198.21 |
| * Portfolio management | 276.19 |

Table 1: Key statistics of FAMMA.

Figure 3: Distribution of questions in FAMMA across languages.

Figure 4: Distribution of questions in FAMMA across subfields.

## 3.1 Dataset Construction

**Question collection.** We assembled a team of seven volunteer STEM researchers to create a comprehensive set of multimodal questions. Five are co-authors of this paper, while the other two are graduates from a Chinese university. Two annotators hold finance degrees, and the others have completed relevant coursework. These annotators draw upon open-source textbooks, exams, and other study materials (see Table 6 in Appendix A for details), and apply their expertise to rewrite or create new questions when needed. The new questions are either entirely original, not present in the data sources, or enhanced versions of existing questions.

The annotators are tasked with selecting questions that require advanced, master-level, or professional knowledge to answer. This selection process is guided by aligning the questions with a minimum of CFA Level 1 difficulty (CFA Institute, 2024b), ensuring they meet industry standards of complexity. Additionally, selected questions must incorporate multimodal information, such as tables, images, or other visual data, to enrich the input and challenge the model's ability to process diverse formats. By following these criteria, we have curated a diverse set of approximately 2,000 questions, drawn from a wide range of authoritative sources.

**Data quality control.** We follow a two-stage data cleaning process to ensure the data quality.

- In the first stage, we conduct a thorough review to correct formatting errors, fix typos, remove duplicate questions, and verify the accuracy of explanations. Each question is cross-verified by 2-3 annotators to ensure consistency and accuracy. Formatting errors and typos arise due to variations in the original sources, such as UTF encoding issues in Chinese and French texts, and the explanations are either provided by the source materials or written by annotators.

- In the second stage, we classify each question into one of three difficulty levels—easy, medium, or hard—and label it with the appropriate subfield.

  The difficulty levels are aligned with the concept-specific standards of the CFA curriculum (CFA Institute, 2024a). In addition, questions that require processing more complex information, such as multiple tables and images, are considered more difficult. In cases where the difficulty is ambiguous, the annotators use their judgment. Additionally, questions deemed overly simplistic—such as those based purely on memorization or with answers that are obvious from the context—are removed to maintain the desired level of challenge and to ensure they test knowledge and reasoning. The subfield annotation is determined by the explicit topics provided in the data source. If the subfield is not clearly specified, the annotators use their discretion to assign the most appropriate category based on the content of the question.

The JSON formats of the multiple-choice and open questions are illustrated in Listing 5 and Listing 6 in Appendix A, respectively.

To conclude, the FAMMA dataset offers a diverse range of questions, enabling the evaluation of models across various scenarios and allowing for fine-grained analysis of their performance.

## 4 EXPERIMENTS

### 4.1 EXPERIMENTAL SETUP

**Benchmarked MLMMs.** We evaluate three cutting-edge closed-source models that are ranked among the top 10 on the Multimodal Arena Leaderboard (Lmsys Org, 2024):

- GPT-4o (OpenAI, 2024a): The latest iteration in the GPT series, GPT-4o features enhanced capabilities in language and vision understanding, as well as improved generation performance.
- Claude-3.5-Sonnet (Anthropic, 2024): Developed by Anthropic, Claude Sonnet introduces architectural innovations that improve multimodal dialogue and reduce harmful outputs.
- Claude-3-Sonnet (Anthropic, 2024): An earlier version of the Claude Sonnet model.

Additionally, we assess a leading open-source model: Qwen2-VL (Yang et al., 2024) that achieves state-of-the-art performance on visual understanding benchmarks, including MathVista (Lu et al., 2024) and DocVQA (Mathew et al., 2021).

**Generation process.** MLMMs are instructed to understand the format and the structure of the questions, and return the response, under a zero-shot setting on our benchmark. The instruction prompts are designed to be straightforward and consistent across all models. Please refer to Listing 7 and Listing 8 in Appendix B for the prompts used to to guide responses to multiple-choice and open questions, respectively. During the final stage of generation for multiple-choice questions, we utilize both regex and GPT-4o to extract the corresponding lettered option from the response. Any discrepancies between the two methods will be manually reviewed and validated by annotators.

**LM-powered evaluation.** During the evaluation process, we use GPT-4o as an LM evaluator to assess the accuracy of responses generated by LLMs for each question. The reported score represents the accuracy of these responses. Each response is categorized as either correct or incorrect, and the reported score reflects the average accuracy across the entire set of questions.

The LM evaluator is instructed to understand the format and structure of the questions, as well as to consider the key points in the ground-truth answers for evaluating the responses. Please refer to Listing 9 in Appendix B for the instructions provided for evaluating the answers. Note that for open-ended questions, where both gold and generated answers are provided, there is a single correct answer, making the 1-0 correctness straightforward to determine. We set the temperature of the LM evaluator as 0 to keep the evaluation results deterministic.

**Human performance.** We invite two volunteers to participate in the test to establish a human benchmark. Both are experienced finance professionals: one holds a Master's degree in Finance from a U.S. institution, while the other graduated from a Grande École in France, specializing in mathematics and finance. The first volunteer, proficient in both English and Chinese, is tasked with completing half of the English test and the entire Chinese test, while the second volunteer takes the remaining portion of the English test and the full French test. They are allowed to consult textbooks, e.g. Hull (2017); Bodie et al. (2014), but are prohibited from searching the web for answers.

It worth noting that the human score is roughly the same as those estimated from CFA passing scores. Based on the report [2], the passing score is approximately $68\%$ for all the three levels. During the annotation process, the difficulty levels of FAMMA's questions—easy, medium, and hard—closely correspond to those of CFA Levels I, II, and III. Based on this data and assumptions about the performance of unqualified candidates from previous levels, we estimate the accuracy rate for easy, medium, and hard questions to be equal to that of Level I, II, III—$68\%, 62.24\%, 57.26\%$, respectively, which resulting a overall score of $62.1\%$. See Appendix B.1 for details of the estimation.

## 4.2 RESULTS AND ANALYSIS

**Main results.** We repeat the generation and evaluation process three times, and report the average result along with the standard error across all experiments. See the overall scores, breakdown by difficulty levels and languages in Table 2. We summarize the key findings as follows.

- FAMMA presents a comprehensive challenge. Human performance sets the highest benchmark with an overall score of 56.96, leading across all difficulty levels. Among the models, GPT-4o ranks first with a score of 42.11, followed closely by Claude-35-Sonnet at 41.87. Both models fall approximately 15 points short of human performance and, based on our estimates, about 20 points below CFA professional levels. This substantial gap underscores the significant challenges FAMMA poses for MLLMs.

- The open-source Qwen2-VL significantly lags behind more advanced closed-source MLLMs. According to its technical report (Wang et al., 2024a), Qwen2-VL has not been explicitly optimized for financial corpora, whereas the Claude family models prioritize finance as a key domain for evaluation and improvement (Claude, 2024). Interestingly, Qwen2-VL performs better on hard questions than on medium ones. A possible explanation is that hard questions often require higher computational complexity and advanced mathematical reasoning, areas where Qwen2-VL excels. In fact, its technical report highlights superior performance on MathVista (Lu et al., 2024), outperforming other MLLMs, including GPT-4o and the Claude models.

To conclude, the main results highlight the progress in MLMM QA in finance but also underscores the challenge of surpassing human-level performance.

**Analysis I: model performance across different subfields.** As shown in Figure 5, GPT-4o demonstrates the largest margin in economics, a social science discipline that studies the behaviour and interactions of economic agents. The result indicates its rich knowledge in social domains in addition to mathematics reasoning. GPT-4o also excels at financial statement analysis, whose context usually contains long tables (see Figure 2a), indicating its superior ability in table understanding (Kim et al., 2024). This well-rounded performance suggests that GPT-4o possesses a broad understanding of diverse financial concepts, excelling in knowledge-based and applied assessments.

Claude-35-Sonnet leads in corporate finance, alternative investments, derivatives, and fixed income, though with small margins over GPT-4o. Both Qwen2 and Claude-3-Sonnet fall significantly short in most areas. The notable improvement in finance-related QA from Claude-3 to Claude-35-Sonnet is consistent with public findings, where the win rate on finance tasks improved by $27\%$, as reported in the technical report (Claude, 2024).

**Analysis II: model performance across different languages.** Seen from Table 2, a consistent observation is that all models perform best in English, with GPT-4o and Claude-35-Sonnet comparably surpass the other competitors with a score around 44. For both GPT-4o and Claude-35-Sonnet, Chinese performance falls noticeably behind English, especially in harder categories, suggesting that

---

[2]https://300hours.com/cfa-passing-score/

| MODELS | SIZE | OVERALL | EASY | MEDIUM | HARD |
|---|---|---|---|---|---|
| HUMAN PERFORMANCE | N/A | 56.16 | 61.35 | 57.09 | 52.11 |
| GPT-4O | N/A | 42.85 | 47.25 | 39.85 | 40.98 |
| | | (0.45) | (4.72) | (1.71) | (4.59) |
| * ENGLISH | | **44.90** | 48.00 | 43.83 | 42.71 |
| | | (0.10) | (4.67) | (0.19) | (4.52) |
| * CHINESE | | **37.70** | 39.45 | 37.35 | 30.65 |
| | | (2.80) | (3.95) | (4.04) | (2.04) |
| * FRENCH | | 32.50 | 64.75 | 31.55 | 22.90 |
| | | (1.65) | (4.17) | (2.63) | (0.73) |
| CLAUDE-35-SONNET | N/A | 42.80 | 47.37 | 41.10 | 39.89 |
| | | (0.49) | (4.12) | (2.72) | (3.89) |
| * ENGLISH | | 44.20 | 47.31 | 46.15 | 40.89 |
| | | (0.35) | (4.47) | (0.39) | (3.52) |
| * CHINESE | | 37.50 | 47.37 | 37.50 | 22.45 |
| | | (2.92) | (3.55) | (4.24) | (2.15) |
| * FRENCH | | **37.50** | 50.00 | 31.58 | 46.88 |
| | | (1.15) | (3.27) | (2.93) | (1.89) |
| QWEN2-VL | 70B | 34.50 | 38.39 | 27.40 | 35.67 |
| | | (0.33) | (2.52) | (3.43) | (4.19) |
| * ENGLISH | | 36.20 | 38.73 | 28.21 | 37.08 |
| | | (0.51) | (4.17) | (1.19) | (3.07) |
| * CHINESE | | 29.60 | 34.21 | 31.25 | 18.37 |
| | | (1.32) | (2.15) | (4.02) | (1.98) |
| * FRENCH | | 25.79 | 50.00 | 18.42 | 34.38 |
| | | (1.85) | (3.25) | (2.68) | (2.19) |
| CLAUDE-3-SONNET | N/A | 31.55 | 31.91 | 29.00 | 32.58 |
| | | (0.34) | (4.42) | (2.22) | (4.08) |
| * ENGLISH | | 32.70 | 33.27 | 31.62 | 32.65 |
| | | (0.28) | (4.17) | (1.39) | (3.82) |
| * CHINESE | | 25.32 | 19.74 | 28.91 | 24.49 |
| | | (3.22) | (4.15) | (3.98) | (2.92) |
| * FRENCH | | 30.53 | 50.00 | 21.05 | 43.75 |
| | | (1.28) | (3.77) | (3.13) | (2.82) |

Table 2: The score of various models on the FAMMA test set, with standard errors indicated in parentheses. The anonymous live-updating leaderboard is available at: https://random2024go.github.io/indexPage/

there are significant gaps in how well these models handle complex tasks in Chinese. The comparatively low scores for Chinese might reflect the challenges related to tokenization or potentially smaller and less diverse training corpora in Chinese relative to English. In addition, Qwen2-VL, perform poorly on Chinese test, indicating it falls short of training on Chinese financial corpus

Claude-35-Sonnet outperforms its competitors in French, likely due to the Claude family's focus on optimizing non-English languages, such as Spanish and French (Claude, 2023; 2024). Interestingly, it achieves a higher score on hard questions compared to medium ones. It's important to highlight that the number of hard questions in French is limited to just 32 (see Table 8 in Appendix A), primar-

ily covering portfolio management, derivatives, and fixed income—three subfields where Claude-35-Sonnet consistently excels with high scores.

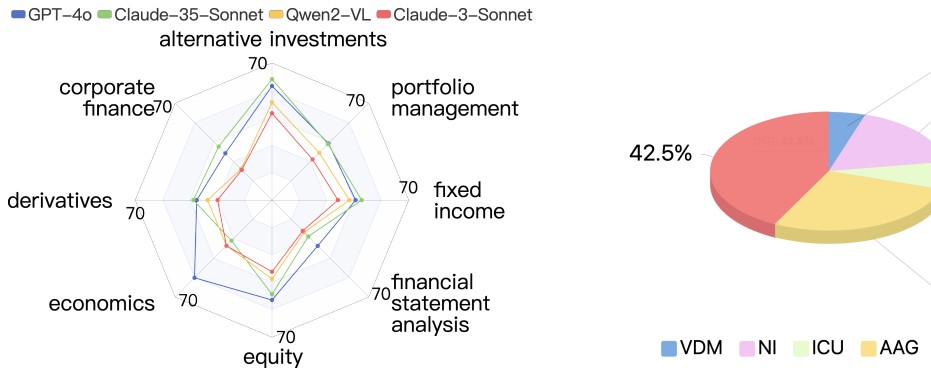

Figure 5: Performance breakdown by subfields in the FAMMA test set.

Figure 6: Distribution of error types of GPT-4o's responses in the sampled set.

Overall, GPT-4o and Claude-35-Sonnet demonstrate robust, well-rounded language skills, while Qwen2-VL and Claude-3 show areas for improvement, particularly in non-English contexts. This suggests that language support and training on diverse financial corpora play a key role in overall model performance in multilingual financial QA tasks.

**Analysis III: error characterisation.** To investigate the limitations of current MLLM capabilities on FAMMA, we conduct a comprehensive analysis of error types observed in our evaluation. We meticulously examined 100 randomly sampled error instances from GPT-4o's generations in a single experiment run. The sample includes 76 responses in English, 14 in Chinese, and 10 in French, with a distribution across difficulty levels of 30 easy, 30 medium, and 40 hard.

Through our analysis, we identified five common error types and categorized all instances accordingly, with detailed examples provided below and in Appendix B.2:

- Data misinterpretation (DM): errors where the model fails to correctly interpret the input data, whether textual, visual (e.g., charts, tables), or a combination of both. For instance, as shown in Listing 11 in Appendix B.2, GPT-4o incorrectly reads a number from a figure, where a small section of the image has low resolution due to the data source.

- Incomplete context understanding (ICU): errors where the model fails to understand the full context of the question or misses critical details. See Listing 12 in Appendix B.2 for an example that GPT-4o overlooks the option and reaches to a wrong result.

- Numerical inaccuracy (NI): errors involving incorrect calculations or misinterpretation of numerical data. Listing 13 in Appendix B.2 presents a response where GPT-4o produces incorrect decimal values during a square calculation.

- Domain knowledge gaps (DKG): errors where the model lacks sufficient understanding of specific financial concepts or practices. See Listing 1 below for an example that GPT-4o misunderstands the nature of the high-touch agency approach in financial markets.

- Ambiguous answer generation (AAG): errors where the model provides vague or incomplete responses that do not fully answer the question, or fails to clearly and correctly align its internal computation with the correct final choice. Listing 2 below shows an example that GPT-4o correctly performs the computation but arrives at an in-correct final result.

**Analysis IV: common error patterns.** Seen from Figure 6, a significant portion, 42.5%, of errors is attributed to DKG, highlighting that GPT-4o struggles to handle financial domain knowledge, which may indicate limitations in its training on financial data. AAG accounts for 17.5% of the errors, indicating that the model sometimes generates unclear or imprecise responses despite correctly understanding the question. NI represents 17.5% of the errors, suggesting that GPT-4o still faces challenges with precise calculations or handling numerical values correctly. The remaining two error types are less frequent but still notable, showing some difficulty in interpreting context or visual

```
⋮
Small currency trades and small
exchange-traded derivatives trades
are typically implemented using the
direct market access (DMA) approach,
and the high-touch agency approach is
typically used to execute large,
non-urgent trades in fixed-income and
exchange-traded derivatives markets.
```

Listing 1: A sample case response of GPT-4o due to domain knowledge gaps: the high-touch agency approach is mainly used for illiquid orders but exchange-traded derivatives tend to be very liquid.

```
⋮
The expected holding-period return (
HPR) is calculated by multiplying
each possible return by its
probability and summing the results.
 The calculation is as follows: (0.30
 * 18%) + (0.50 * 12%) + (0.20 * -5%)
 = 5.4% + 6% - 1% = 10.4%. Therefore,
 the expected HPR for KMP stock is
 10.88%.
```

Listing 2: A sample case response of GPT-4o due to ambiguous answer generation: it correctly performs the computation but reaches to a wrong final result.

data. These findings underscore the need for improvements in finance domain-specific training, and answer generation to enhance GPT-4o's performance.

| | TOP 1 ERR | TOP 2 ERR |
|---|---|---|
| ENGLISH | DKG (32.65%) | AAG (25.45%) |
| CHINESE | DKG (44.16%) | AAG (34.08%) |
| FRENCH | DKG (46.08%) | AAG (39.04%) |

Table 3: Error types breakdown by language in the sampled set.

| | TOP 1 ERR | TOP 2 ERR |
|---|---|---|
| EASY | AAG (34.33%) | DKG (32.20%) |
| MEDIUM | AAG (36.49%) | DKG (30.12%) |
| HARD | DKG (57.08%) | AAG (26.12%) |

Table 4: Error types breakdown by difficulty in the sampled set.

**Analysis V: errors across languages and difficulties.** In all languages, DKG consistently account for the highest proportion of errors, with French leading at $46.08\%$, indicating GPT-4o struggles with understanding finance domain-specific knowledge, particularly in non-English contexts. Interestingly, AAG error emerges as the second most common error type across all three languages, suggesting that despite differences in language complexity, GPT-4o often provides unclear or incomplete answers regardless of the language.

When analyzing error types across difficulty levels, AAG dominates in both easy ($34.33\%$) and medium ($36.49\%$) categories, while DKG takes the lead in hard questions ($57.08\%$). This shift suggests that for easier questions, the model tends to generate ambiguous answers, likely due to overgeneralization or incomplete interpretations. However, as the complexity increases, the GPT-4o's lack of domain knowledge becomes more evident.

**Analysis VI: can RAG or o1-reasoning chain help?** We explore two independent methods for improving GPT-4o's performance on FAMMA:

- Retrieval augmented generation (RAG): we augment GPT-4o with external financial knowledge base by incorporating content from textbooks "CFA Level III SchweserNotes, Books 1-5, 2023", which comprehensively cover most of the topics included in FAMMA.

- Dynamic Chain-of-Thought (COT) prompting: we implement o1-style reasoning chains (OpenAI, 2024b), where at each step, GPT-4o can either proceed to the next reasoning step (by trying multiple methods, exploring alternative answers, or questioning previous solutions) or provide a final answer. The process begins with a system prompt that includes instructions to guide a step-by-step reasoning approach. Once the problem is introduced as a user message, an assistant message is pre-loaded to establish a standardized starting point for the GPT-4o's response generation.

See Appendix B.3 for the implementation of RAG and dynamic COT (prompts in Listing 16- Listing 18). These two strategies are applied on the same error set again. Table 5 shows dynamic COT significantly outperforms RAG, correcting 44 errors compared to RAG's 20. It is particularly effective in the AAG category and also excels in NI and ICU corrections. Listing 4 presents an

example of the reasoning process, involving identification, calculation, and self-checking, to ultimately arrive at an answer. Listing 3 illustrates the final answer generation, clearly reaching the correct result instead of producing an ambiguous response. This highlights dynamic COT's ability to resolve unclear answers and enhance numerical reasoning. As expected, RAG performs better at addressing DKG errors, though both methods show limited effectiveness in correcting VDM.

|  | RAG | DYNAMIC COT |
|---|---|---|
| # CORRECTED | 20 | 44 |
| - DKG | 10 | 8 |
| - AAG | 5 | 20 |
| - NI | 8 | 13 |
| - VDM | 0 | 0 |
| - ICU | 1 | 3 |

Table 5: Improvement by applying RAG and dynamic COT in the sampled set.

```
The table provided gives us the
probabilities and holding-period
returns (HPR) for KMP stock under
different economic scenarios: Boom,
Normal Growth, and Recession.
The expected holding-period return (
HPR) can be calculated using the
formula for expected value:

⋮

Boom: Probability = 0.30, HPR = 18%\
n2. Normal Growth: Probability =
0.50, HPR = 12%\n3. Recession:
Probability = 0.20, HPR = -5%.

To find the expected HPR:\nE(HPR) =
(0.30 * 18%) + (0.50 * 12%) + (0.20
* -5%)\n= (0.30 * 18/100) + (0.50 *
12/100) + (0.20 * -5/100)\n= 0.054 +
0.06 - 0.01\n= 0.104 or 10.40%
```

Listing 3: A corrected AAG case using dynamic COT, as shown in Listing 2: no additional incorrect results were generated.

```
{
    "title": "Understanding the
Given Probability Distribution",
    "content": "The table provided
gives us the probabilities and
holding-period returns (HPR) for KMP
 stock under different economic
scenarios: Boom, Normal Growth, and
Recession.",
    "next_action": "continue"
},
{
    "title": "Calculating the
Expected Holding-Period Return",
    "content": "The expected holding
-period return (HPR) can be
calculated by ...(omitted)",
    "next_action": "continue"
},
{
    "title": "Performing the
Expected Value Calculation",
    "content": "To find the expected
 HPR:...(omitted)",
    "next_action": "continue"
},
{
    "title": "Identifying the
Correct Answer",
    "content": "Based on..., the
expected holding-period return for
KMP stock is 10.40%.",
    "next_action": "final_answer"
}
```

Listing 4: Reasoning process using dynamaic COT.

## 5 CONCLUSION

In this paper, we introduced a comprehensive benchmark for multilingual multimodal QA within the financial domain, addressing the growing need for robust systems that can interpret and respond to queries in various languages while leveraging diverse data modalities.

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

| MATERIALS | LANGUAGE | SOURCE |
|---|---|---|
| QUIZZES ON FINANCE-RELATED COURSES | ENGLISH | MIT OPEN-COURSE |
| TEXTBOOK: NUMERICAL PROBABILITY | ENGLISH | ONLINE PDF |
| TEXTBOOK: PAUL WILMOTT ON QUANTITATIVE FINANCE | ENGLISH | GITHUB |
| QUIZZES ON FINANCE-RELATED COURSES | CHINESE | RENMING UNIVERSITY |
| QUIZZES ON FINANCE-RELATED COURSES | FRENCH | SCRIBD |
| QUIZZES ON FINANCE-RELATED COURSES | FRENCH | ACADEMIA |

Table 6: Selected sources as references for generating questions.

# Appendices

## A    DATASET DETAILS

**Data source.**    The question-response pairs are primarily collected from free online resources, quizzes, textbooks, and other study materials. See Table 6 for more details.

**Data format.**    Following data validation, we provide the following information for each question:

- Question ID: a unique identifier for the question acroos the whole dataset.
- Context: relevant background information related to the question.
- Question: the specific query being asked.
- Images: directories of images referenced in the context or question.
- Options: a list of possible answers, applicable only to multiple-choice questions.
- Question type: categorized as either multiple-choice or open-ended.
- Main question ID: a unique identifier for the question within its context; questions with the same context share the same ID.
- Sub question ID: a unique identifier for the question within its corresponding main question.
- Answer: a concise and accurate response.
- Explanation: a detailed justification for the answer.
- Images for explanation: directories of images supporting the explanation.
- Subfield: the specific area of expertise to which the question belongs, categorized into eight sub-fields.
- Language: the language in which the question text is written.
- Difficulty: a measure of the question's complexity based on the level of reasoning required.

## B    EXPERIMENT DETAILS

### B.1    HUMAN PERFORMANCE ESTIMATION

Based on the analysis of CFA exams (see https://300hours.com/cfa-passing-score/), the passing score is approximately $68\%$ for all the three levels. During the annotation process, the difficulty levels of FAMMA's questions—easy, medium, and hard—closely correspond to those of CFA Levels I, II, and III. For the medium questions, we assume those who fails the Level I will have a score of $50\%$ on Level II, therefore the corresponding score for Level II over the whole population is $68\% * 68\% + (1 - 68\%) * 50\% = 62.24\%$. By similarly assuming those who are not qualified for

| Statistics | Numbers (Percentage) |
|---|---|
| Total Questions | 1758 |
| *  MC | 976  (55.5%) |
| *  Open | 782  (44.5%) |
| *  w. explanations | 1750  (99.5%) |
| *  w. multiple images | 521  (29.6%) |
| Total Subfields | 8 |
| *  Alternative Investments | 87 (4.9%) |
| *  Corporate Finance | 256 (14.6%) |
| *  Derivatives | 254 (14.4%) |
| *  Economics | 33 (1.9%) |
| *  Equity | 256 (14.6%) |
| *  Fixed Income | 95 (5.4%) |
| *  Financial Statement Analysis | 248 (14.1%) |
| *  Portfolio Management | 529 (30.1%) |
| Total image types | 3 |
| *  Questions w. tables | 1426 (81.1%) |
| *  Questions w. charts | 278 (15.8%) |
| *  Questions w. screenshots | 54 (3.1%) |
| Difficulties | 608 / 438 / 712 |
| (E: M: H) | 34.6% : 24.9% : 40.5% |
| Splits | 120 / 1638 |
| (Validation:Test) | 6.8% : 93.2% |
| Languages | 1385 / 253 / 120 |
| (English:Chinese:French) | 78.8% : 14.4% : 6.8% |
| Avg length in tokens | |
| *  Questions | 233.43 |
| *  Explanation | 73.95 |

Table 7: More detailed key statistics of FAMMA.

Level III will have a score of $40\%$ on Level III, the expected score of the whole population for Level III becomes $62.24\% * 68\% + (1 - 62.24\%) * 30\% = 57.26\%$. In this context, we set the human score for easy, medium, and hard questions to be equal to that of Level I, II, III—$68\%, 62.24\%, 57.26\%$, respectively, which resulting a overall score of $59.86\%$.

## B.2 Case Studies

We present a few sample cases of error response from GPT-4o.

• Data misinterpretation: GPT-4o incorrectly reads a figure as $16\%$ instead of the correct value of $18\%$, likely due to a small section of the figure having low resolution from the data source, which leads to inaccurate calculations (see Listing 11).

• Incomplete context understanding: GPT-4o overlooks the option "one of the options are correct" when generating a response, resulting in an incorrect answer (see Listing 12).

• Numerical inaccuracy: GPT-4o produces incorrect decimal values during a square calculation, leading to an erroneous output (see Listing 13).

| SUBFIELD | ENGLISH | CHINESE | FRENCH |
|---|---|---|---|
| ALTERNATIVE INVESTMENTS | 27 / 17 / 39 | 3 / 1 / 0 | - |
| CORPORATE FINANCE | 72 / 61 / 50 | 1 / 35 / 1 | 0 / 33 / 3 |
| DERIVATIVES | 37 / 24 / 172 | 9 / 5 / 0 | 0 / 0 / 7 |
| ECONOMICS | 10 / 1 / 21 | 1 / 0 / 0 | - |
| EQUITY | 85 / 24 / 74 | 20 / 13 / 7 | 2 / 28 / 3 |
| FINANCIAL STATEMENT ANALYSIS | 56 / 21 / 18 | - | - |
| FIXED INCOME | 76 / 27 / 82 | 11 / 22 / 12 | 7 / 7 / 4 |
| PORTFOLIO MANAGEMENT | 157 / 59 / 175 | 31 / 52 / 29 | 3 / 8 / 15 |
| TOTAL | 520 / 234 / 631 | 76 / 128 / 49 | 12 / 76 / 32 |
| | (34.5%:13.4%:52.1%) | (27.7%:50.6%:21.7%) | (7.5%:59.2%:33.3%) |

Table 8: Distribution of questions in difficulty across languages and subfields in FAMMA.

- Domain knowledge gaps: GPT-4o misunderstands the nature of the high-touch agency approach in financial markets, confusing its application in exchange-traded derivatives and large trades (see Listing 14).
- Ambiguous answer generation: GPT-4o correctly performs the computation but arrives at an incorrect final result due to ambiguity in answer interpretation (see Listing 15).

These instances are firstly categorized by LM-evaluators (see Listing 10 in Appendix B for the instruction prompt), then validated by human expert based on their knowledge and the golden explanations if available.

### B.3 DETAILS ON RAG AND o1-REASONING EXPERIMENTS

**RAG setup.** We utilize 5 CFA Level III curriculum textbooks—"CFA Level III SchweserNotes, Books 1-5, 2023", which comprehensively cover most of the topics found in FAMMA— as the external knowledge source. The notes are in PDF format, each consisting of 200-300 pages with quizzes at the end of every chapter, though these quizzes are not included in FAMMA. We upload them to GPT-4o via the API for queries.

**Dynamic COT setup.** The implementation is based on the open source project https://github.com/bklieger-groq/g1, which is originally built on Llama-3.1. We improve the project to be compatible with GPT-4o. The process begins with a system prompt that includes instructions to guide a step-by-step reasoning approach. Once the problem is introduced as a user message, an assistant message is pre-loaded to establish a standardized starting point for the GPT-4o's response generation.

```
{
    "question_id": "
English_validation_86",
    "context": "The following data are
available relating to the performance
of Wildcat Fund and the market
portfolio: <image_1>",
    "question": "The risk-free return
during the sample period was 7%.
Calculate Sharpe's measure of
performance for Wildcat Fund.",
    "options": "['1.00%', '8.80%',
'44.00%', '50.00%']",
    "image_1": "/9j/4
AAQSkZJRgABAQAAAQABAAD...]",
    "image_2": null,
    "image_3": null,
    "image_4": null,
    "image_5": null,
    "image_6": null,
    "image_7": null,
    "image_type": "table",
    "answers": "C",
    "explanation": "(18 - 7)/25 =
.44.",
    "topic_difficulty": "easy",
    "question_type": "multiple-choice",
    "subfield": "portfolio management",
    "language": "english",
    "main_question_id": 369,
    "sub_question_id": 2,
    "ans_image_1": null,
    "ans_image_2": null,
    "ans_image_3": null
}
```

Listing 5: Multi-choice questions in JSON representation.

```
{
    "question_id": "
English_validation_42",
    "context": "Cleveland
Compressor and Pnew York Pneumatic are
competing manufacturing firms. Their
financial statements are printed here
.<image_1><image_2><image_3><image_4
>",
    "question": "Which firm has the
larger investment in current assets?
Why?",
    "options": "",
    "image_1": "/9j/4
AAQSkZJRgABAQAAAQABAAD...",
    "image_2": "/9j/4
AAQSkZJRgABAQAAAQABAAD...",
    "image_3": "/9j/4
AAQSkZJRgABAQAAAQABAAD...",
    "image_4": "/9j/4
AAQSkZJRgABAQAAAQABAAD...",
    "image_5": null,
    "image_6": null,
    "image_7": null,
    "image_type": "table",
    "answers": "Cleveland
Compressor.",
    "explanation": "Cleveland
Compressor holds the larger investment
in current assets. It has current
assets of $92,616 while Pnew York
Pneumatic has $70,101 in current assets
. The main reason for the difference is
 the larger sales of Cleveland
Compressor.",
    "topic_difficulty": "hard",
    "question_type": "open question
",
    "subfield": "financial
statement analysis",
    "language": "english",
    "main_question_id": 329,
    "sub_question_id": 3,
    "ans_image_1": null,
    "ans_image_2": null,
    "ans_image_3": null
},
```

Listing 6: Open questions in JSON representation.

```
You are a highly knowledgeable
financial expert. Please answer
multiple-choice questions in the
finance domain. You are given context,
images, questions and options.
The questions are multilingual (either
in English, Chinese, or French) and
multimodal (containing images as part
of the question). '<image_1>, <image_2
> ...' mentioned in the text of the
context or question are sequential
placeholders for images, which are fed
at the same time as the textual
information.
If no image information is provided,
you must answer based solely on the
given information.
Besides, the question may contain
several sub-questions that share the
same context, and the answer to each
sub-question may depend on the answers
to previous ones.
The question format is

context: <context>
sub-question-1: <sub-question-1>
sub-question-2: <sub-question-2>
sub-question-3: <sub-question-3>
...

Now consider the following question:
context: {context}
{sub_questions}

Please provide the chosen answer and a
precise, detailed explanation of why
this choice is correct. The explanation
 should be in the same language as the
question and should not exceed 400
words.
Your response must be in a standard
JSON format:
{{
    sub-question-1: {{
        answer-1: <answer-1>,
        explanation-1: <explanation-1>
    }},
    sub-question-2: {{
        answer-2: <answer-2>,
        explanation-2: <explanation-2>
    }},
    sub-question-3: {{
        answer-3: <answer-3>,
        explanation-3: <explanation-3>
    }},
    ...
}}
Ensure that the response strictly
adheres to JSON syntax without any
additional content.
```

Listing 7: Format of our instruction prompt on multi-choice questions.

```
You are a highly knowledgeable
financial expert. Please answer open-
ended questions in the finance domain.
The questions are multilingual (either
in English, Chinese, or French) and
multimodal (containing images as part
of the question). '<image_1>, <image_2
> ...' mentioned in the text of the
context or question are sequential
placeholders for images, which are fed
at the same time as the textual
information.
If no image information is provided,
you must answer based solely on the
given information.
Besides, the question may contain
several sub-questions that share the
same context, and the answer to each
sub-question may depend on the answers
to previous ones.
The question format is

context: <context>
sub-question-1: <sub-question-1>
sub-question-2: <sub-question-2>
sub-question-3: <sub-question-3>
...

Now consider the following question:
context: {context}
{sub_questions}

Please provide the answer and a precise
, detailed explanation. The explanation
 should be in the same language as the
question and should not exceed 400
words.
Your answer must be in a standard JSON
format:
{{
    sub-question-1: {{
        answer-1: "answer-1",
        explanation-1: "explanation-1"
    }},
    sub-question-2: {{
        answer-2: "answer-2",
        explanation-2: "explanation-2"
    }},
    sub-question-3: {{
        answer-3: "answer-3",
        explanation-3: "explanation-3"
    }},
    ...
}}
Ensure that the response strictly
adheres to JSON syntax without any
additional content.
```

Listing 8: Format of our instruction prompt on open questions.

```
You are a highly knowledgeable expert
and teacher in the finance domain.
You are reviewing a student's answers
to financial questions.
The questions are multilingual (either
in English, Chinese, or French) and
multimodal (containing images as part
of the question). '<image_1>, <image_2
> ...' mentioned in the text of the
context or question are sequential
placeholders for images, which are fed
at the same time as the textual
information.
You are given the context, the question
, the student's answer and the student'
s explanation and the ground-truth
answer.
Please use the given information and
refer to the ground-truth answer to
determine if the student's answer is
correct.

The input information is as follows:

context: {context}
question: {question}
student's answer: {model_answer}
student's explanation: {
model_explanation}
ground-truth answer: {answer}

Please respond directly as either '
correct' or 'incorrect'.
```

Listing 9: Format of our prompt on judging the correctness of the model output.

```
You are a highly skilled expert in
error analysis for AI models in the
finance domain. You are reviewing
collected incorrect answers to
financial questions.
The questions are multilingual (either
in English, Chinese, or French) and
multimodal (containing images as part
of the question). '<image_1>, <image_2
> ...' mentioned in the text of the
context or question are sequential
placeholders for images, which are fed
at the same time as the textual
information.
You are given the context, the question
, the student's answer, the student's
explanation and the ground-truth.

You need to classify these incorrect
answers based on the provided
categories: perceptual errors, lack of
knowledge, reasoning errors, and other
errors. Here are the definitions for
each error type:

Data misinterpretation: .(omitted)
Incomplete context understanding: ...(
omitted)
Numerical inaccuracy:  ...(omitted)
Domain knowledge gaps:  ...(omitted)
Ambiguous answer generation:  ...(
omitted)

The input is as follows; use these
details to determine the primary error
category.

context: {context}
question: {question}
model incorrect answer: {model_answer}
model explanation: {model_explanation}
ground-truth answer: {answer}

Now please provide the result directly,
 identifying the error category as one
of: data misinterpretation, incomplete
context understanding, numerical
inaccuracy, domain knowledge gaps, or
ambiguous answer generation.
```

Listing 10: Format of our prompt on error analysis on model's output.

```
⋮
 The expected return (E[R]) is
calculated as follows:
E[R] = (0.30 * 16%) + (0.50 * 12%) +
(0.20 * -5%) = 4.8% + 6% - 1% = 9.8%
```

Listing 11: A sample error response of GPT-4o due to data misinterpretation: it incorrectly reads the figure as 16% instead of the correct value of 18%.

```
⋮
Therefore, the expected holding-period
return for the stock is 10.4%. However,
since 10.4% is not one of the given
options, the closest correct answer is
8.33%, which is option B.
```

Listing 12: A sample error response of GPT-4o due to incomplete context understanding: it overlooks the option of "None of the options are correct".

```
⋮
For Normal growth: 0.45 * (15% -
14.35%)^2 = 0.45 * (0.65%)^2 = 0.45 *
0.004225 =  0.00190125
```

Listing 13: A sample response of GPT-4o due to numerical inaccuracy: the square calculation results in incorrect decimal values.

```
⋮
Small currency trades and small
exchange-traded derivatives trades are
typically implemented using the direct
market access (DMA) approach, and the
high-touch agency approach is typically
used to execute large, non-urgent
trades in fixed-income and
exchange-traded derivatives markets.
```

Listing 14: A sample case response of GPT-4o due to domain knowledge gaps: the high-touch agency approach is mainly used for illiquid orders but exchange-traded derivatives tend to be very liquid.

```
⋮
 The calculation is as follows: (0.30 *
 18%) + (0.50 * 12%) + (0.20 * -5%) =
5.4% + 6% - 1% = 10.4%. Therefore, the
expected HPR for KMP stock is 10.88%.
```

Listing 15: A sample case response of GPT-4o due to ambiguous answer generation: it correctly performs the computation but reaches to a wrong final result.

```
{
    "role": "system",
    "content": """You are an expert AI
assistant who explains your reasoning
process step by step. For each step,
provide a title describing what you're
doing in that step, and the content.
Determine whether another step is
needed or if you're ready to give a
final answer. Respond in JSON format
with 'title', 'content', and '
next_action' (which can be 'continue'
or 'final_answer') keys. Use multiple
reasoning steps whenever possible, at
least 3. Be aware of your limitations
as an LLM and what you can and cannot
do. In your reasoning, include
exploration of alternative answers.
Consider that you might be wrong and
where errors in your reasoning might
occur. Thoroughly test all other
possibilities. You may be wrong. When
you say you're revisiting, actually
revisit and use a different method to
do so. Don't just say you're revisiting
. Use at least 3 methods to arrive at
an answer. Use best practices.

    Example of a valid JSON response:
    ```json
    {
        "title": "Identifying Key
Information",
        "content": "To begin solving
this problem, we need to carefully
examine the given information and
identify the key elements that will
guide our solution process. This
involves...",
        "next_action": "continue"
    }```
"""
}
```

Listing 16: Format of the system prompt used in dynamic COT.

```
{
    "role": "assistant",
    "content": "Thank you! I will now
think step by step following my
instructions, starting at the beginning
 after decomposing the problem."
}
```

Listing 17: Format of the assistant prompt used in dynamic COT.

```
{
    "role": "user",
    "content": "Please provide the
final answer based on the above
reasoning.",
}
```

Listing 18: Format of the user prompt used in dynamic COT.

