# OpenReview forum: "FAMMA: A Benchmark for Financial Multilingual Multimodal Question Answering"
_ICLR.cc/2025/Conference — Submitted to ICLR 2025_

### Official Review · Reviewer_6rqb · 2024-11-01

**Soundness:** 3
**Presentation:** 4
**Contribution:** 3
**Rating:** 6
**Confidence:** 4

**Summary:**

This paper introduces FAMMA, a benchmark for financial question answering, which includes 1) multimodal and multilingual input contexts and 2) challenging questions that are difficult even for finance master’s students. Evaluation results show that recent LLMs struggle with this dataset, with GPT-4 achieving only 42.85% accuracy. A case analysis reveals that LLMs mainly fail due to a lack of domain-specific knowledge and ambiguous answer generation.

**Strengths:**

1. Evaluating and benchmarking LLMs is crucial, especially since many existing benchmarks are becoming saturated. This paper introduces a challenging benchmark specifically in the finance domain, which could facilitate the deployment of LLMs in the financial industry.
2. The data is sourced from high-quality class quizzes and has been verified by humans, enhancing its reliability.
3. The results highlight that recent powerful LLMs still struggle significantly with this task.

**Weaknesses:**

1. The paper does not address the possibility of data contamination, considering that some questions may have been sourced from online resources that might appear in the LLM's pretraining corpus.
2. The authors could have evaluated additional inference-time methods, such as self-consistency and program-of-thoughts.
3. The number of questions in Chinese and French is relatively small, which limits the interpretability of results for these languages. Using GPT-4o to translate English questions into Chinese and French (with human verification) can augment the size of multilingual data.
4. The paper lacks an ablation study on the importance of multimodal input. It would be beneficial to show the performance of text-only input and examine the impact of displaying tables as images versus converting them into text.

**Questions:**

1. Can you provide the distribution of different modalities (e.g., charts, tables, diagrams)?
2. For RAG or Dynamic CoT, you show they can correct certain errors, which is promising. However, do these methods negatively impact the performance on questions that the vanilla approach already answers correctly?
3. How does the o1-preview model perform?
4. Are there any copyright concerns associated with publicly releasing these questions?

**Details Of Ethics Concerns:**

The dataset includes questions sourced from finance quizzes in university courses and finance textbooks worldwide, which may raise copyright issues. The authors do not address this concern in the paper.

---

### Official Review · Reviewer_EEq2 · 2024-11-04

**Soundness:** 2
**Presentation:** 3
**Contribution:** 3
**Rating:** 6
**Confidence:** 4

**Summary:**

This paper introduces a new benchmark for question answering in the financial domain called FAMMA. FAMMA is created from publicly available finance courses and other resources and is annotated by experts in the field. It includes 1,758 questions and answers. Just over half of the questions are multiple-choice, while the rest have very short answers, often a number.

Notable features of this dataset, compared to existing ones, include:
1. Requiring a high level of expertise, for example, "master's level knowledge of statistics and optimization theory."
2. Requiring understanding of tables, charts, diagrams, and text
3. Coverage of three languages: English, Chinese, and French.

The paper then evaluates several state-of-the-art vision and language models on this dataset, including GPT-4o, two versions of Claude-Sonnet, and the open model Qwen2-VL-70B.
It also experiments with retrieval-augmented generation and Chain-of-Thought prompting with GPT-4o, which outperform simple prompting.

The authors conclude that there is a large gap between human performance and all evaluated LLMs on FAMMA. They perform error analysis on the outputs of GPT-4o and identify several categories of errors, including a gap in knowledge about the financial domain, and numerical inaccuracies.

**Strengths:**

The process of sourcing questions for FAMMA appears thorough, and the annotations provided by domain experts are valuable. The paper includes an excellent error analysis section that enhances the understanding of the challenges involved.

The evaluation is conducted on a well-balanced mix of closed and open-source vision and language models, and average of three runs is reported. This strengthens the conclusion that LLMs fall short on this task.

**Weaknesses:**

1. The relatively low human performance (L270), as stated in the paper, raises concerns about the quality of the dataset. What are the reasons for this relatively low human performance? Are the test-takers not familiar with all of the subdomains? Is there any ambiguity in the questions? How might this affect the label quality, which is also determined by human annotators? What is the inter-annotator agreement rate when "each question is cross-verified by 2-3 annotators" (L218)?

2. The results suggest that Chain-of-Thought (CoT) prompting outperforms humans. According to Table 5, CoT corrects 44 errors (out of 100). Does this imply that it would score approximately `100 - [(100 - 42.85) * (1 - 0.44)] = 68%` if run on the entire test set, surpassing the human performance of 56% from Table 2? If my understanding is correct, this would undermine the claim that FAMMA poses a significant challenge for LLMs, and would put the benefits of this new dataset into question. In any case, I recommend reporting the full performance of RAG and CoT on the entire test set for comparison, instead of running them on just a subset of the errors.

3. Additionally, the details of the RAG setup are confusing. What embedding model is being used for retrieval, and what is the prompt used for generation? Is OpenAI’s commercial RAG offering being used? If so, there are more performant retrieval models in the literature, which may close the gap between RAG and CoT.

Other Minor Issues:

1. Short-form answers make automatic evaluation easier. However, what if a model arrives at the right answer for the wrong reasons, especially in multiple-choice questions? It might be worth considering requiring models to show their work for some of the questions.

2. "o1-style reasoning" (line 476) should cite [1] as well. The implementation seems to be a modified version of Chain-of-Thought as it was originally proposed in [1], not an "internalized" Chain-of-Thought as in OpenAI’s o1.

3. Please reduce the empty space in tables.

**Reference:**

1. Chain-of-Thought Prompting Elicits Reasoning in Large Language Models

**Questions:**

1. In LLM-powered evaluation (L261): How much does it differ from a simple EM metric, given that most answers are either multiple choice or a short answer?
2. What version of GPT-4 is used for the experiments (what date)?
3. What is the rationale behind the heavily unbalanced data split between the validation and test sets?

---

### Official Review · Reviewer_zM3M · 2024-11-05

**Soundness:** 2
**Presentation:** 3
**Contribution:** 3
**Rating:** 5
**Confidence:** 4

**Summary:**

The paper introduces FAMMA, an open-source benchmark designed for evaluating financial multilingual multimodal question-answering (QA) capabilities in large language models. FAMMA includes 1,758 question-answer pairs derived from university finance textbooks and exams, covering eight key finance subfields like corporate finance, asset management, and financial engineering. These questions are presented in multiple languages (mainly English, but also Chinese and French) and combine text with various image types, such as charts and tables. Evaluation results show that current multimodal models struggle with FAMMA’s challenges; even advanced models like GPT-4o achieve only 42% accuracy, significantly below human-level performance.

**Strengths:**

Novelty and Relevance: FAMMA provides a timely and innovative benchmark, specifically tailored to the financial domain with a unique focus on multilingual and multimodal question-answering.

**Weaknesses:**

1. Data Quality Assurance: The paper lacks quantitative analysis on data quality control, which raises concerns about the reliability and consistency of the benchmark. A more detailed breakdown of the two-stage data cleaning process, including quantitative results from the initial data cleaning phase and the difficulty-level annotation phase, would improve clarity on data preparation and quality control.

2. Copyright and Usage Rights: A significant concern is potential copyright issues with the data sources, as all data appears to be derived from textbooks, exams, and other educational materials that may have restricted usage for research. The paper should explicitly address these concerns by providing evidence of permissions or free-use provisions to ensure transparency and compliance with copyright regulations.

3. Model Evaluation Scope: The evaluation includes a limited set of multimodal LLMs, mainly focusing on a few proprietary models and open-source model. Expanding the model set to include a broader range of multimodal LLMs would offer a more comprehensive assessment of FAMMA’s benchmarking capabilities and further validate the benchmark’s utility across diverse model architectures.

**Questions:**

See weakness

**Details Of Ethics Concerns:**

A significant concern is potential copyright issues with the data sources, as all data appears to be derived from textbooks, exams, and other educational materials that may have restricted usage for research. The paper should explicitly address these concerns by providing evidence of permissions or free-use provisions to ensure transparency and compliance with copyright regulations.

---

### Meta-Review · Area_Chair_iqE2 · 2024-12-17

**Metareview:**

This paper introduces a new dataset, FAMMA, for multilingual multimodal question answering (QA) in finance. Several MLLMs are evaluated on the dataset and the results show FAMMA poses a significant challenge for MLLMs. The dataset may be beneficial to a narrow group of the community and its impact may be limited. The experimental analysis is generally comprehensive.  The reviewers raised several major issues regarding the dataset, evaluation and presentation. First, the dataset may suffer from data contamination issues. I think that direct verification using LLMs in the author's rebuttal does not address this issue well. Second, the number of questions/contexts in some language is very small. As a multilingual dataset, it may be more useful to have enough questions and contexts in different languages. The presentation of this paper needs substantial improvement, e.g., the details of data quality control need to be added in the next version. In summary, this paper is a borderline paper. I have mixed feelings about it, a bit negative.

**Additional Comments On Reviewer Discussion:**

Two reviewers participated in the discussion and slightly increased their scores. Reviewer zM3M didn't engage in the discussion.

---

### Decision · Program_Chairs · 2025-01-22

Reject